

# Proximate composition, functional properties and quantitative analysis of benzoyl peroxide and benzoic acid in wheat flour samples: effect on wheat flour quality

Numrah Nisar[1], Faiza Mustafa[1], Arifa Tahir[1], Rashad Qadri[2], Yaodong Yang[3], Muhammad Imran Khan[4] and Fuyou Wang[3]

[1] Department of Environmental Sciences, Lahore College for Women University, Lahore, Punjab, Pakistan
[2] Institute of Horticultural Sciences, University of Agriculture Faisalabad, Faisalabad, Punjab, Pakistan
[3] Hainan Key Laboratory of Tropical Oil Crops Biology/Coconut Research Institute, Chinese Academy of Tropical Agricultural Science, Wenchang, Hainan, China
[4] Institute of Soil and Environmental Sciences, University of Agriculture Faisalabad, Faislabad, Punjab, Pakistan

Corresponding author
Rashad Qadri,
rashad.qadri@uaf.edu.pk

## ABSTRACT

**Background:** Extensive milling processes have deprived wheat flour from essential nutrients. The objective of the current study was to assess the nutritive quality of commercial wheat flour (soft flour (SF)) through analyses of proximate composition and functional properties as well as quantification of benzoyl peroxide (BPO; added as bleaching agent in the SF) by comparing the results with whole wheat flour (WF; never received any additives).

**Methods:** The samples included commercial SF purchased from the local supplier of different flour mills (who use BPO as additive) and a control sample without additives was prepared by grinding the seeds harvested from wheat (*Triticum aestivum* L.; Inqulab 91) crop grown in the experimental field of University of Agriculture, Faisalabad, under optimized field conditions without any fertilizers and insecticides. Functional properties (including bulk density, water absorption capacity, oil absorption capacity, emulsifying activity, foaming capacity, least gelatinization concentration and gelatinization temperature) and proximate composition (including moisture content, ash contents, crude protein, gluten and starch contents) were determined and compared for all the samples. Benzoyl peroxide (BPO) and Benzoic Acid (BA) quantification was performed through High Performance Liquid Chromatography. Finally dietary intake was estimated for BPO and BA.

**Results:** Results showed that SF had lesser fiber, protein and ash contents, whereas, higher damaged starch, fat, gluten and bulk density. A parallel experiment under selected conditions (temperature, time and solute concentration) showed dissociation of BPO into BA soon after the exposure. Observed BA range (13.77 mg/g after 16 h) in SF and exposure level assessment (44.3 ± 1.36 mg/kg/BW) showed higher intake of BA on the consumption of SF. The results revealed the superiority of WF over SF in nutritive qualities as well as free of toxicants such as BA.

# INTRODUCTION

Wheat (*Triticum aestivum* L.) is a principal cereal consumed worldwide in different forms as a major food source. During milling process the grains are milled to obtain flour which is pale yellow in color. The bakery products it yields are less commercially acceptable, therefore flour is bleached for different bakery items. When this flour is stored, natural maturation or aging takes place, where carotenoids undergo oxidation and improves rheological properties of the dough such as texture (*Liu et al., 2014*). Lutein (a xanthophyll) is excessively found in the wheat seeds (≈3 mg/kg of dried wheat) and the yellow color of the bread is due to the remains of lutein in wheat seeds (*Mellado-Ortega & Hornero-Méndez, 2017*).

Wheat milling industries utilize oxidizing agents such as benzoyl peroxide (BPO) to accelerate the process of maturation and dough improvement. These agents oxidize sulf-hydryl groups in flour gluten protein that yields increased stickiness of the flour and makes its appearance as soft flour (SF) (*Onishi et al., 2004*). These oxidizing agents can be used for bleaching and dough improvement. During flour bleaching conjugated double bond of carotenoids is disrupted to less conjugated colorless system, which gives flour a desirable texture for baking. One of the most commonly used oxidizing agent is BPO which exhibits bleaching properties without influencing the baking or taste (*Gaddipati, Volpe & Anthony, 1983*). BPO is a free radical initiator and it produces carotenoid oxidation like a typical free radical mechanism. No acceptable limits of BPO have been specified in the food safety regulations. BPO has been extensively used throughout the world as a bleaching agent without any recovery. In food processing BPO is (>92%) converted into benzoic acid (BA), which is usually used as a preservative for soft drinks, fruits, juices and many other types of food items. The higher concentration of BA than allowable safety level (40 mg/kg) is harmful for humans. Acute toxicity is unknown, however, a sensitive person consuming BA more than 5 mg/kg of body weight (BW) per day can initiate non-immunological (pseudo-allergy) reactions, hyperpnea, metabolic acidosis and convulsions (*Wei, Zhujun & Liu, 2006*; *Liu, 2007*).

Beside addition of BPO, the SF is processed several times compared to the WF to achieve the fiber free fine particles for the improved texture of the bakery products. During this processing structure, sensory qualities, protein contents, fiber contents and nutritional components get severely affected. Antioxidants in WF, which are present in the germ and bran (*Isabelle & Andre, 2006*), are mostly removed in the refined flour. Therefore, the current study was designed to observe and compare the proximate composition and functional properties of SF and WF as well as BPO and BA concentrations in them. Current study also considered the conversion of BPO to BA at different time intervals after adding a known concentration of BPO to WF (control; without additives; grown under similar conditions to control with additives). The findings will help to identify the levels of

BA in SF which may get consumed and nutritional deprivation as a result of intensive grinding and sieving of wheat flour during milling.

## MATERIALS AND METHODS

### Sample collection

Test samples included commercial SF ($n = 4$, SF) purchased from the local supplier from different flour mills (with additives) and a control sample ($n = 1$, WF) without additives was prepared by mechanical grinding the *Triticum aestivum* L. seeds (Inqulab 19), harvested from the crop grown on the experimental field of University of Agriculture, Faisalabad, under optimized field conditions (16/8 D/N; 23 ± 1 °C; 14 Inch water) without any fertilizer and insecticide. The variety of wheat seeds and growth conditions were kept same as used by flour mills to generate SF, to avoid ambiguity in results. The WF flour sample was passed through sieve (75 μm size) before packing them into air-tight plastic containers. The size was in agreement with the particle size of SF purchased from flour mills.

### Determination of functional properties

Bulk density of flour was determined as described previously (*Lam et al., 2008*) by following ASTM D1895B prescribed procedure. The sample was allowed to flow freely in a circular container (0.615 L) with a suspended funnel of opening diameter (1.5 cm). The height of funnel was kept about 20 cm and the powder was stirred continuously to avoid clogging inside the opening. Container with the sample was dropped few times from the height of 150 mm to allow settling and release of air. Weight of the container with the sample was determined and weight/volume (loose bulk density) was obtained. Density was determined through the formula, $d = \dfrac{m}{v}$, where mass ($g$) is the weight of the sample and volume (mL) is the volume of the material. Water absorption capacity and oil absorption capacity (WAC and OAC respectively) were determined through the method described by *Beuchat (1977)*. In this method, 1 g of sample was allowed to mix with 10 mL of distilled water for about 30 s. Sample was then allowed to stand at room temperature (25 ± 2 °C) for the next 30 min and centrifuged at 3,000 rpm (30 min). Volume of the supernatant was determined and WAC (mg/mL) was calculated by formula $WAC = V_{initial} - V_{final}$, where $V$ is the volume of water (mL). Similar procedure was repeated for OAC determination using soybean oil (Sp. Gravity 0.9092) as the absorbing agent. Emulsifying stability and emulsifying activity (ES and EA respectively) were determined by the method described by *Neto et al. (2001)* with modifications. About five mL of flour dispersion (10 mg/mL of water) was homogenized with five mL of soybean oil for 1 min through vigorous shaking. The emulsion was then centrifuged (2–6; Sigma, Neustadt, Germany) at 1,100 rpm for 10 min. Height (cm) of emulsified layer (ELH) was deducted from the total height of the tube contents (TC) to estimate the EA by the equation $EA = \left(\dfrac{ELH}{TC}\right) \times 100$. ES was calculated by heating the emulsion at 80 °C for 30 min before centrifuging at 1,300 rpm for 10 min. ES was then calculated by $ES = \left(\dfrac{ELHA}{TCA}\right) \times 100$, where ELHA is the height of emulsified layer after heating and TCA is total content of the tube before heating. The effect of EA and ES was determined by varying

the capacity of the flour samples. Foaming stability and capacity (FS and FC) of flours were determined by the method reported by *Coffmann & Garciaj (1977)*, where about 100 mL of distilled water was mixed with 10 g of flour. The suspension was mixed vigorously for 5 min on the magnetic stirrer (MSC Digital, IRMCO, Berlin, Germany). The initial solution volume $V_1$ and the final solution volume $V_2$ were recorded (using graduated cylinder). Foaming Capacity (FC) was also calculated from the formula $FC = \left(\frac{V_2 - V_1}{V_1}\right) \times 100$. Foaming Stability (FS) was also determined by the foam volume that was left for 8 h and expressed as percentage of initial foam volume. To determine the gelation properties; Least gelatinization concentration (LGC) and gelatinization temperature (GnT), distilled water sample suspensions (2–10% w/v) were prepared. About 10 mL of these suspensions were transferred into the test tubes. In the boiling water bath these test tubes were heated for 1 h and then cooled for 2 h at 4 °C in a refrigerator. Least gelation concentrations were taken when the samples did not fall from the inverted test tubes.

## Proximate composition

Proximate composition was determined through FT-NIR spectroscopy (TENSOR 37 FTIR Spectrometer; Burker, Bremen, Germany) as well as conventional methods to get reliable results. These properties included moisture content (MC), ash contents (AC), crude protein (CP), gluten and starch contents. Other properties such as crude fiber and fat (CFF) were determined through solvent extractions. For crude fiber about 2.5–3 g of sample was transferred to Soxhlet and extracted with petroleum ether. The air dried extracted sample was transferred to one L conical flask. Whole boiling dilute sulfuric acid (1.25% w/v) was transferred to the flask containing sample and immediately connected with a water cooled reflux condenser and heat it so that the contents start boiling in 1 min. Few drops of octanol were added to avoid bumping and risk of foaming. The flask was continuously shaken and boiled for 30 min. Flask is then removed and filtered through fine linen (18 threads to a cm) placed in a funnel and washed with boiling water until no longer acid to litmus is achieved. The residue on linen was then washed with the 200 mL of boiling NaOH (1.25% w/v). Immediately the flask was connected to the reflux condenser and boiled for 30 min. The residue was then filtered through filtering cloth and residues were thoroughly washed with boiling water. The residues were transferred to Gooch crucible having thin compact layer of ignited asbestos. Finally wash the residues with 15 mL ethyl alcohol. The Gooch crucible was dried in air oven at 105 ± 2 °C in an air oven until constant weight is achieved. The crucible was weighed and placed in a muffle furnace to burnt all carbonaceous matter. Crucible was then cooled and weighed again. Crude fiber was then calculated by the following formula; crude fiber % by wt $= \frac{(W_1 - W_2) \times 100}{W}$, where $W_1$ is weight of Gooch crucible before ashing, $W_2$ is weight of Gooch Crucible containing asbestos and ash and $W$ is weight of the dried material taken for the test. The fat content of wheat was determined using the method of Association of Official Analytical chemists (*AOAC International, 2000*) with petroleum ether as solvent. In this method the fat was extracted by petroleum ether.

Sample of about 3.0 g was weighed and added in a labeled thimble. About 150 mL of petroleum ether was then added to it and boiled up to 40–60 °C in 250 mL boiling flask. The thimble was tightly plugged with cotton wool and Soxhlet apparatus was assembled to allow it to reflux for 24 h the thimble was carefully removed and petroleum ether from the top was drained into another container for further use. The petroleum ether was then evaporated completely in a hot air oven and desiccator was weighed following cooling. The weight was determined by formula; fat content % by wt $= \frac{W_1 \times 100}{W}$, where $W_1$ is the weight of the residue in desiccator and $W$ is initial weight of the sample.

Since the bulk density (BD; ρ) varies with MC, therefore, it was determined following procedure described in ASABE standard S358.2 (*Theerarattananoon et al., 2011*). During this procedure, 100 g of sample was dried in a forced air convection oven (IM-115, Berlin, Germany) at 103 °C for 24 h. The sample was then weighed on digital balance (0.01 g precision; TE-313S-DS, Göettingen, Germany) and MC was calculated by the equation; $MC = W_{initial} - W_{final}$. MC was also determined through FT-NIR spectroscopy. Flour samples of different moisture level were utilized to develop the model for calibration and multivariate analysis was conducted after gathering their spectra. Unknown samples were then analyzed against the calibration curve to get the moisture contents. Crude protein was determined through semi micro-Kjeldahl method (AACC adopted method 46-13; *AACC, 1995*) and auto protein analyzer (Kjeltec 2400 auto-analyzer; FOSS Analytical, Hillerød, Denmark). Here 1 g of flour sample was used along with keeping nitrogen to protein conversion factor of 5.7. AACC method 38-12 was utilized to determine the gluten contents of the selected flour samples (25 g each).

## Analytical method

NIR Omega G Analyser (Bruins Instruments, Salem, NH, USA) was employed to analyze different parameters (protein, starch, fat, moisture and gluten) of the flour and grain samples. The spectral transmissions range was 700–1,100 nm with 5 nm scan increment, measured at controlled room conditions of 24 ± 1 °C, RH 34 ± 2%.

## MIR spectroscopy

The FTIR transmission spectra were recorded at Burker-TENSOR 37 FTIR spectrometer with Michelson interferometer. Working range of the spectrometer was 4,000–12,000 cm$^{-1}$ and spectra generated were interpreted on the basis overtones of different functional groups in the product. Resolution of spectrometer was 4 cm$^{-1}$ (max scan interval value was 2 cm$^{-1}$) with maximum scan time kept at 5 s. MIR spectra were recorded at Opus 6.0 Burker Software using Attenuated Total Reflectance (ATR) unit. The reference spectrum (empty sample bottle) was utilized as background measurement before loading in sample's spectra. About 8–10 g of sample was added in the sample bottle to generate the spectra in diffused reflectance mode. Three spectra per sample were recorded by rotating the sample bottle at 120°. The measurements were carried out under the controlled conditions of temperature 24 ± 1 °C, RH 34 ± 2%. Each spectra were the average of three scans per object.

## BP and BA quantification

### Sample preparation and bleaching reaction

Benzoyl peroxide and BA were quantified in wheat flour as described by the method of (*Onishi et al., 2004*). Whole wheat sample was taken and divided into two parts; one sample was kept as control and BPO was added in the other. About 50 g of flour was blended with 1.5 mg of BPO (bleaching agent) to achieve a concentration of 30 µg/g. The mixture was passed through polyester sieve (400 mesh/cm$^2$) to attain the homogeneous blend and kept in dark. The bleaching reaction (performed at room temperature) was monitored after every hour. The product of reaction was extracted every hour for a period of 8 h, finally a sample was taken at 12 h and 16 h. Then samples were analyzed by High Performance Liquid Chromatography (HPLC) and compared with the commercial standards and calibration graph for the quantification. Associated peaks (matching with the standard) were considered for the generation of results.

## Calibration graph for BPO

Standard stock solution (SS) was generated by dissolving pure BPO (60 mg/L) in diethyl ether (100% purity). Working standards were obtained by diluting SS with appropriate volume of diethyl ether. For BA stock solution, pure compound (100 mg) was dissolved in 100 mL of methanol and working standards were prepared by diluting the stock solution. Calibration curve was then generated by plotting the absorbance value against concentration.

## Extraction procedure for BPO and BA

The standard procedure was carried out (at room temperature) in a flask with grinding stopper. About 100 mL of diethyl ether was added to 50 g of flour (both control and the one which is already mixed and sifted with BPO). This mixture was shaken vigorously on a magnetic stirrer for 10 min and left to settle for 15 min. Upper layer of this solution (containing the products of reaction) was withdrawn through the pipette and transferred into Falcon polypropylene tube (10 mL) and held into ice until HPLC analysis.

## HPLC method

The supernatant was analyzed by Waters 600 HPLC system at Inertsil ODS-80A column (5 µm, 4.6 × 250 mm; GL Science, Tokyo, Japan) equipped Inertsil ODS-3 guard column (10 × 4 mm i.d.) and Waters 2,996 Photodiode array detector. The detection wavelength was kept at 235 nm and column oven at 40 °C. For isocratic separation the conditions were as follows: Water (Solvent A), acetonitrile (CAN; solvent B) and Benzoic acid (Solvent C); 55% B:45% A as mobile phase for 1 mL/min. The gradient conditions for analysis were as follows: Water-glacial acetic acid (1,000:1) (Solvent A), ACN-glacial acetic acid (1,000:1) (solvent B); 18% B (10 min) was increased to 60% B (11–15 min hold) at flow rate 1.2 mL/min and column temperature 35 °C.

### Estimation of dietary intake

The mean dietary intake for both BP and BA were estimated to determine the exposure rate. For this purpose, 200 subjects (35–40 years of age group; random sampling) were

evaluated for their preferences of WF and SF as well as amount of daily intake through a questionnaire survey. The SF brands, which were reported to be consumed, were actually evaluated for the presence of BA in the current study. Further calculations were accomplished through the following equation:

$$Y = \sum (X_v \times C_v)/B_w$$

where, $X_v$ = average daily amount (kg) of wheat flour consumed by a subject; $C_v$ = concentration of BA (mg/kg) as determined through HPLC in particular wheat flour sample; $B_w$ = Average body weight of the subject.

## Statistical analysis

All samples were analyzed in triplicate (both biological and experimental replicates), therefore standard error of mean (SEM) was applied using the Statistixl 1.9 Add-in package within Excel 2007. Two-way analysis of variance (ANOVA) was conducted, followed by post-hoc Tukey tests that separated the treatment into groups (at $P < 0.01$ and $P < 0.05$). The aim was to give the significant difference in the data sets from different flour samples which was not achievable through univariate or one-way ANOVA.

# RESULTS

Both the commercially available SF and whole wheat flour (WF) were compared for their functional properties, emulsifying properties, foaming capacity, gelation capacity, proximate composition and benzoyl peroxide composition. Finally, the exposure of benzoyl peroxide was compared with the daily intake capacity to observe the exposure of consumers when they are consuming SF or WF. The samples of SF were collected from the flour mills which were preferably consumed by the subjects. WF samples were collected from same seed variety (Inqulab 19) grown under similar conditions as wheat of SF. Detailed results are as under.

## Functional properties

The oil absorption capacity (OAC; Table 1) showed that the SF sample has highest lipophilic tendency of about 2.87 mL/g. Highest OAC (188 mL/100 g compared to 146 mL/100 g for WF) and WAC (408 g/100 mL compared to 140 g/100 mL for WF) were obtained for SF4. All considered flour mill samples had almost similar results for OAC. Water Absorption Capacity (WAC) was, however, higher (140 g/100 mL) for WF compared to SF (<123 g/100 mL).

The results revealed that emulsifying Activity (EA) was higher for WF (43.7%), whereas, stability (ES) is higher for all SF samples (<42%). EA and ES of the WF and SF may also vary due to the milling process. The emulsifying properties varied inversely, which means that WF had highest EA and lowest ES. Foaming Capacity (FC) and Foaming stability (FS) collectively form the foaming properties of any flour. Both of these properties are directly proportional to one another, which were observed to be higher for WF (12.9% FC and 1.94% FS). FC and FS of WF is more (<12% and <1% respectively) compared

**Table 1 Functional properties of the whole wheat flour (WF) and soft flour (SF) samples.**

| Samples | SC at 95 °C (g/mL) Mean ± SD | WAC (g/100 mL) Mean ± SD | OAC (g/100 g) Mean ± SD | EA (%) Mean ± SD | ES (%) Mean ± SD | FC (%) Mean ± SD | FS (%) Mean ± SD | GnT (°C) Mean ± SD | LGC (%) Mean ± SD | BD (ρ) Mean ± SD |
|---|---|---|---|---|---|---|---|---|---|---|
| WF | 17.8 ± 1.2 | 140 ± 5.77 | 146 ± 9.33 | 43.7 ± 2.61 | 38.4 ± 5.77 | 12.9 ± 1.02 | 1.94 ± 0.07 | 59.21 ± 1.39 | 8 ± 1.65 | 0.76 ± 0.048 |
| SF1 | 19.4 ± 0.98 | 121 ± 4.04 | 181 ± 8.75 | 37.84 ± 3.47 | 44.6 ± 7.54 | 8.97 ± 1.22** | 0.77 ± 0.065 | 41.21 ± 1.66* | 6 ± 0.98* | 0.23 ± 0.06 |
| SF2 | 18.8 ± 0.97 | 123 ± 8.93 | 175 ± 7.63 | 36.44 ± 7.12 | 45.4 ± 8.75 | 8.66 ± 1.08** | 0.84 ± 0.078 | 38.7 ± 1.86* | 4 ± 0.87** | 0.24 ± 0.056 |
| SF3 | 19.6 ± 0.81 | 120 ± 2.36 | 184 ± 8.55 | 38.45 ± 3.22 | 43.5 ± 8.41 | 8.71 ± 1.76** | 0.76 ± 0.065 | 37.74 ± 2.06* | 4 ± 0.65** | 0.21 ± 0.081 |
| SF4 | 19.7 ± 1.72 | 118 ± 3.45 | 188 ± 9.34 | 36.7 ± 4.21 | 42.9 ± 7.66 | 8.34 ± 0.51** | 0.74 ± 0.045 | 38.45 ± 2.76* | 4 ± 0.77** | 0.26 ± 0.056 |

Notes:
* $P < 0.05$, significant difference as calculated through ANOVA.
** $P < 0.01$, significant difference as calculated through ANOVA.
SC, swelling capacity; WAC, water absorption capacity; OAC, oil absorption capacity; EA, emulsifying activity; ES, emulsifying stability; FC, foaming capacity; FS, foaming stability; GnT, gelatinization temp.; LGC, least gelatinization concentration; BD, bulk density.

**Table 2 Proximate composition of whole wheat flour (WF) and soft flour (SF) samples.**

| Samples | Moisture (%) Mean ± SD | Ash (%) Mean ± SD | Crude fiber (%) Mean ± SD | Fat (%) Mean ± SD | Crude protein (%) Mean ± SD | Gluten (%) Mean ± SD | Starch (%) Mean ± SD | Sugar (%) Mean ± SD | Damaged starch (%) Mean ± SD | Pentosan (%) Mean ± SD |
|---|---|---|---|---|---|---|---|---|---|---|
| WF | 8.64 ± 0.054 | 1.6 ± 0.08 | 1.44 ± 0.076 | 2.29 ± 0.15 | 8.9 ± 1.24 | 14.4 ± 0.78 | 76.92 ± 4.32 | 16.92 ± 1.07 | 45.84 ± 5.42 | 1.6 ± 0.07 |
| SF1 | 4.25 ± 1.23* | 0.54 ± 0.064 | 0.34 ± 0.076* | 1.54 ± 0.98 | 4.65 ± 0.86** | 7 ± 1.23** | 50.26 ± 1.86** | 23.41 ± 3.74 | 88.36 ± 4.32** | 0.25 ± 0.054 |
| SF2 | 4.22 ± 0.83* | 0.55 ± 0.043 | 0.34 ± 0.07* | 1.32 ± 0.05 | 4.32 ± 0.85** | 7.5 ± 2.21** | 50.4 ± 2.77** | 23.15 ± 8.75 | 87.22 ± 2.05** | 0.23 ± 0.055 |
| SF3 | 3.84 ± 0.67** | 0.45 ± 0.081 | 0.32 ± 0.05* | 1.12 ± 0.063 | 4.15 ± 1.65** | 6 ± 1.22** | 50.21 ± 1.24** | 23.4 ± 6.34 | 82.3 ± 1.24** | 0.24 ± 0.07 |
| SF4 | 3.9 ± 0.17** | 0.34 ± 0.049 | 0.33 ± 0.074* | 1.12 ± 0.048 | 4.12 ± 0.08** | 6.5 ± 1.07** | 50.21 ± 4.32** | 23.25 ± 3.45 | 85.62 ± 2.23** | 0.26 ± 0.05 |

Notes:
* $P < 0.05$, significant difference as calculated through ANOVA.
** $P < 0.01$, significant difference as calculated through ANOVA.

to all SF samples (>9% and >1% respectively). A highly significant difference ($P < 0.01$) was observed when values were compared statistically with those of WF.

## Gelation capacity

Gelation capacity (including gelatinization temperature GnT and least Gelatinization concentration LGC) are attributed and controlled by the balance between hydrophilic interactions and repulsive electrostatic interactions between the water/oil and proteins (*Casanova et al., 2008*). Results (Table 1) indicated that WF had higher gelation capacity (GnT = 59.21 °C and LGC = 8%) compared to all SF samples ($P < 0.001$ when datasets were compared with the WF dataset). It can also be observed that both considered parameters for gelation capacity are directly related to each other such that increase in one also showed increase in the other.

## Proximate composition

The proximate composition (moisture, crude fiber, fat, ash, starch and damaged starch) of all the flour samples is as summarized in the Table 2. Moisture content of SF is less (3.84–4.25%), protein contents of WF were higher (8.9% compared to 4.6% for SF) and total starch was also high for WF (76.92% compared to 50.21% for SF). The results
**Table 3 Recoveries of benzoyl peroxide (BP) and benzoic acid (BA) from the flour.**

| Components | Added amount (μg/g) | Recovery (%) | |
|---|---|---|---|
| | | Gradient (Mean ± SD) | Isocratic (Mean ± SD) |
| Benzoyl peroxide | 7 | 98.1 ± 0.55 | 99.2 ± 0.41 |
| | 30 | 99.5 ± 0.61 | 99.3 ± 0.55 |
| | 60 | 96 ± 0.25 | 99.3 ± 0.21 |
| Benzoic acid | 5 | 91.3 ± 0.44 | – |
| | 10 | 91.3 ± 0.31 | – |

indicated that intense milling processes have detrimental effect on several properties of the flour. Most of the components such as crude protein, gluten, damaged starch datasets for SF showed highly significant difference ($P < 0.01$) when compared with the WF dataset. This showed that quality of flour was deteriorated while processing and refining.

## Benzoyl peroxide concentration

Benzoyl peroxide and BA were determined simultaneously using gradient analysis (Table 3). The retention times were observed to be 17.5 min for BP and 7.8 min for BA (Figs. 1 and 2). The maximum absorption of BPO was obtained at 195 nm and 235 nm, however, a wavelength of 235 nm was kept as standard for measuring the BP in the samples, considering the possible interference with the food ingredients appearing at 195 nm. The calibration curve had excellent linearity for the range of 0.05–16 μg/g for BPO and 0.2–15 μg/g for BA. The acquisition following isocratic gradient with 55% ACN, showed an excellent linearity for BP, however BA was not detected. Therefore, measurements at gradient conditions were kept as standard for the analysis.

Benzoyl peroxide is a free radical initiator and it causes the oxidation of carotenoids by free radical mechanism. The process (Fig. 3) leads to the formation of benzoic acid (BA) as a by-product (*Saiz, Manrique & Fritz, 2001*; *Shan et al., 2007*; *Sumnu & Sahin, 2008*). No BA or BP were observed for the WF sample (Fig. 2). In the controlled samples, 30 μg/g of BP was added to the wheat flour samples (grown under standard field conditions) and 99.5% recovery (29.5 μg/g) was observed soon after adding. After 3 h of bleaching, the amount of BP reduced significantly to 4 μg/g (Fig. 4) that reached to zero after 8 h of exposure. The contents of BA were observed to be 2.84 μg/g as recovered soon after the addition. This quantity increased to 8.9 μg/g after 3 h and increased further to 13.5 μg/g after 8 h of exposure. The contents of BA were determined again after 12 h though very small increase in quantity was observed (13.75 μg/g), which shows that the process of conversion got stabilized with slight variation. To confirm this, the flour was analyzed again after 16 h and quantity of BA didn't change much (13.77 μg/g). Local standards for maximum acceptable quantity have not yet been specified however, international standards (60 μg/g for BP; 0–5 mg/kg for BA as per body weight by JECFA acceptable daily intakes) were considered in the current study. Analysis of WF and SF available

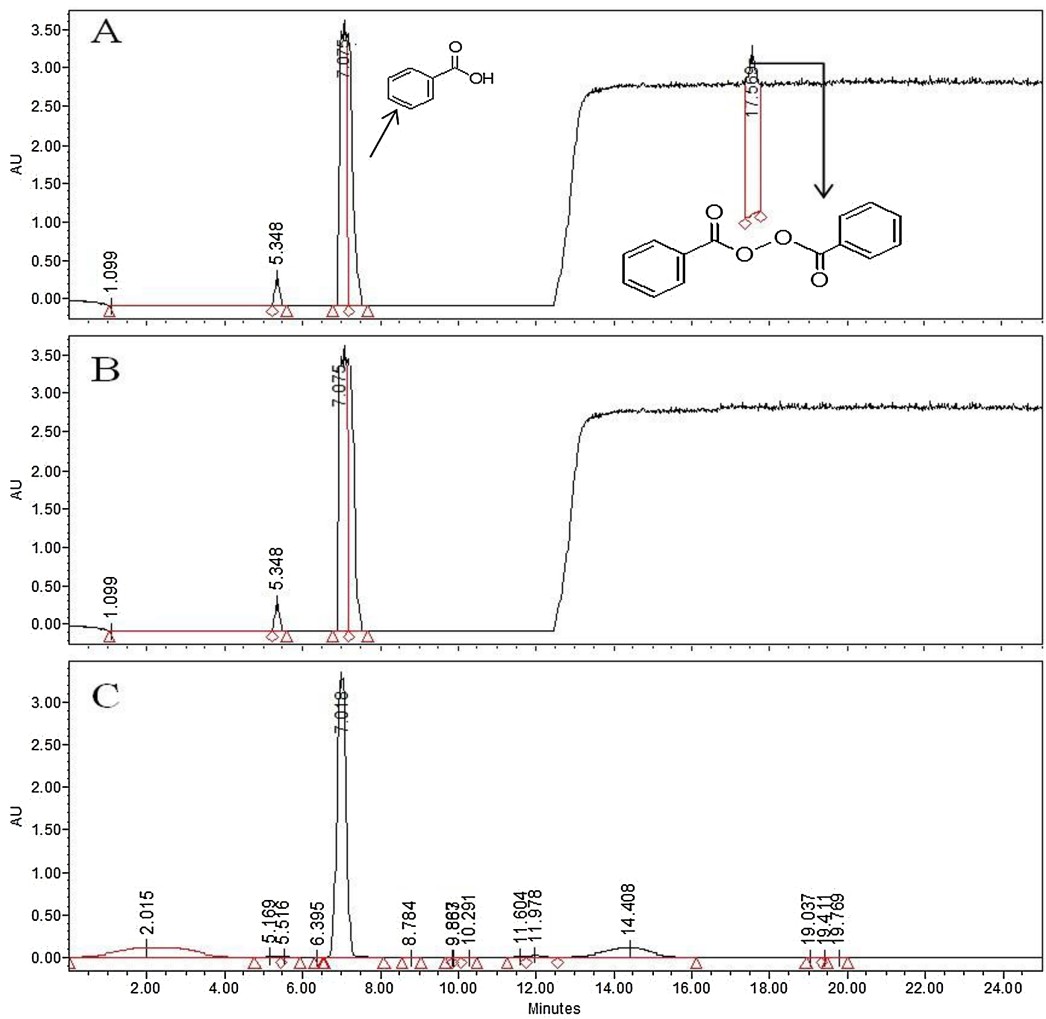

**Figure 1 HPLC chromatogram of BPO (17.56 min) and BA (7.018 min) in WF when induced with 30 µg/g of BP.** Conversion of BP to BA (A) after 2 hrs (B) after 6 hrs (C) after 10 hrs of exposure.

commercially for BP and BA showed higher rates of BA in the SF samples (Table 4). This indicated higher amount of BA intake when SF based products are consumed.

## Estimation of daily intake

Estimation of dietary exposure of BA through just wheat flour consumption was estimated (Table 5). Since flour is important parameter of people diet, therefore, it was deemed necessary to give the level of exposure through flour only. Rest of the food groups such as noodles and drinks, though unavoidable, also contain certain amount of BA but they were not considered in the current study. As evaluated through questionnaire study most of the considered subjects (72.5) were consuming SF purchased from the flour mills.

High Performance Liquid Chromatography was used for quantification of BP in both white and WF samples. BA was also determined because BP was decomposed into benzoic acid within limited days (Fig. 4). A research on bleaching agents including

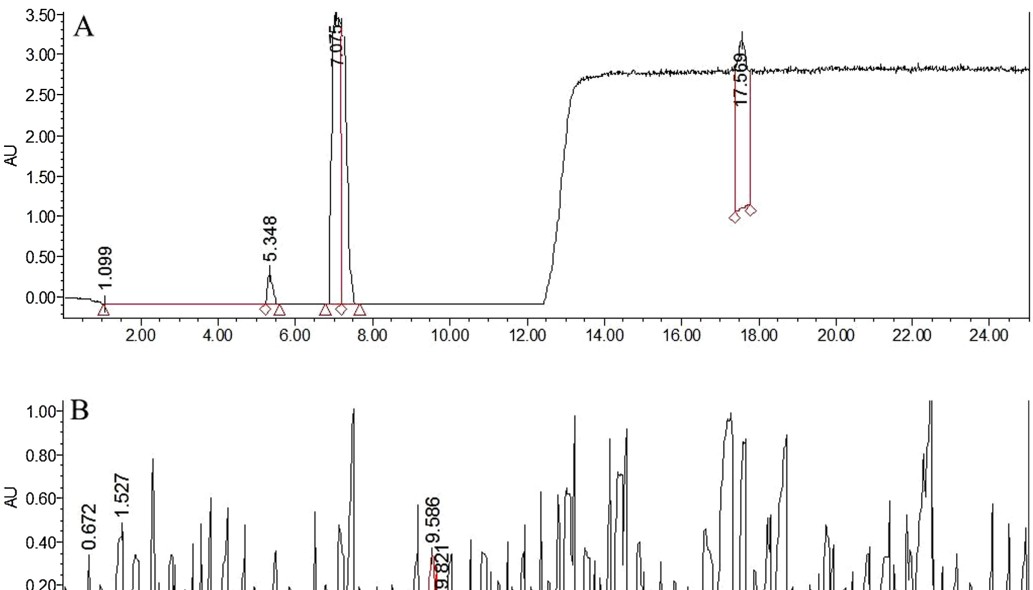

**Figure 2 HPLC chromatograms for BPO and BA.** HPLC chromatograms for BPO and BA in (A) commercial soft flour (SF1) (B) whole flour (WF) without any addition of preservatives.

**Figure 3 Possible pathway for the degradation of benzoyl peroxide (BPO) to benzoic acid (BA) as observed through current research.**

BPO and BA by HPLC during bleaching process of wheat flour. The retention time of BP was 17.5 min and that of BA was 7.6 min. After 30 h of bleaching BPO concentration was 11 ppm. After 3 months its concentration was reduced to 6 ppm. These results demonstrated that when benzoyl peroxide added to flour their greater amount was decomposed into benzoic acid within limited days of treatment. The analytical results of present study showed that that retention time of BPO was 17.5 min and BA was 7.5 min at 235 nm. In WF samples no content of BPO and BA were found. In white flour samples BP content ranges from 6.6 to 21 mg/kg and BA content ranges from 13 to 28 mg/kg.

## DISCUSSION

Whole wheat flour and SF were compared for their properties such as functional properties, emulsifying properties, foaming capacity, gelation capacity, proximate

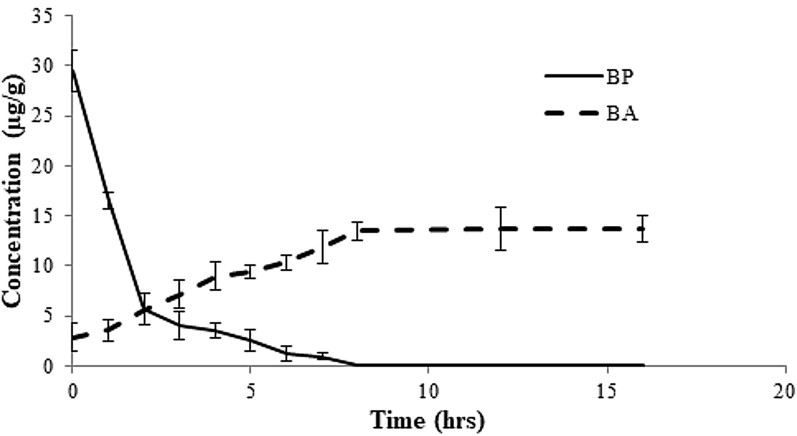

**Figure 4 Rate of conversion of benzoyl peroxide (BP) to benzoic acid (BA) in flour at varied time intervals.** About 50 g of flour was amended with 30 µg/g of BPO as bleaching agent. Rate of dissociation was observed over a period of 16 h.

**Table 4 Concentrations of BP and BA in different flour samples.**

| Samples | Contents (µg/g) | |
|---|---|---|
| | BP (Mean ± SD) | BA (Mean ± SD) |
| WF | 0 | 0 |
| SF1 | 2.45 ± 1.1 | 68.11 ± 14.1 |
| SF2 | 3.41 ± 1.02 | 71.4 ± 12.31 |
| SF3 | 2.54 ± 0.94 | 71.51 ± 15.84 |
| SF4 | 5.77 ± 0.33 | 72.55 ± 15.33 |

Note:
WF, whole wheat flour; SF, soft flour.

**Table 5 Estimated intake of BA on the basis of intake of flour by consumers.**

| Food group | Consumer (%) | Avg. daily intake of flour (g/day) Mean ± SD | Estimated daily intake of BA (mg/kg/bw) Mean ± SD |
|---|---|---|---|
| SF | 72.5 | 78.3 ± 2.3 | 44.3 ± 1.36 |
| WF | 10.5 | 72.1 ± 3.3 | 1.34 ± 1.45 |
| Alternate | 17 | 68.3 ± 1.23 | – |

Note:
Allowable EDI for BA = 0–5 mg/kg/bw.

composition and benzoyl peroxide composition. The comparison was developed to aware and understand the choice of flour in daily meals and their potential associated risks.

The oil absorption capacity (OAC) of the flour varies with the intrinsic properties such as amino acid composition, protein conformation, hydrophilic-hydrophobic balance of amino acids, steric factors as well as lipid and carbohydrate composition of a flour sample (*Mao & Hua, 2012*). "Results" showed that SF samples had more non-polar side chains compared to WF which enhanced the oil binding capacity of SF and reduced its water binding capacity. Higher OAC (188% compared to 146% for WF; Table 1) represents that

the flour can retain flavor and could have optimum uses in different food products such as bakery items. More hydrophobic sites as in case of SF (OAC > 175%), as represented through OAC value, are important for bakery items in which fat absorption is desirable (*Chandra, Singh & Kumari, 2015*; *Chassagne-Berces et al., 2011*). Water Absorption Capacity (WAC) is important since it gives the capacity of a flour to have higher hydration capacity, lower WAC (<123%) in case of SF and higher in case of WF (140%) means that excessive grinding in the flour mills and sieving has resulted in the modification of protein structure. Another factor of higher WAC could be the fibers retaining the water. Reduction of theses fibers in case of SF has reduced its capacity to absorb water (*Onipe, Beswa & Jideani, 2017*; *Shewry, 2009*). WAC is an important component since it allows the food to have sufficient water retention and transfer of this water upon consumption. The interaction of protein with water is usually determined through water hydration, holding, water retention and water imbibing. WAC favors another phenomenon called Water Hydration Capacity (WHC). WHC is a physical feature and describes the ability of flour structure to prevent water from being released from a protein structure. During food processing the protein structures are deteriorated which may influence the WHC of wheat as in case of SF.

Swelling, another important functional property referred to as spontaneous uptake of water by protein matrix, is indirectly related to WHC. Since SF is lower in WHC therefore its swelling capacity is more (>18 compared to 17.8 in case of WF), which ensures longer life of bakery items. Water retention is very important for protein functionality which determines the cationic, non-ionic and anionic polar sites of the protein molecules on the flour surface (*Zayas, 1997*). Lower WHC means that proteins have less water binding capacity or they are "salted out" that takes place when the proteins are precipitated out. *Ramaswamy et al. (2013)* showed that the higher water retention is associated with linear form of arabinan compared to its branched form. The branched form of arabinan is unhealthy for consumption, therefore WF should be the choice for consumption instead of SF. The wheat kernel is usually composed of 12–16% protein, during milling process the protein content and wheat gluten are separated, though gluten is insoluble content of protein. The wheat gluten is further damaged, which may be the reason that in the current results (Table 2) gluten was observed to be higher in concentration compared to protein, since the milling and other processes separated it out.

*Ali et al. (2014)* reported that lesser water retention properties are directly related to the damaged starch contents of the wheat. During milling process, the flour obtained comes from the endosperm which is rich in starch contents. The physical process of grinding during which the cylinders move closer and closer causes the starch granules to rupture. This results in the damaged starch contents and also has detrimental effect on the protein composition. The higher water retention reflects the absorption of water in the starch granules, which also limit its propagation. In case of WF, the flour is produced through grinding the whole seed (endosperm and periphery) which is rich in pentosan, starch, ash and protein in more or less original form.

The emulsifying properties vary inversely with the insoluble protein fractions and polysaccharides (*Chove, Grandison & Lewis, 2001*; *Haruna, Udobi & Ndife, 2011*).

The unfolding of proteins when at interface water/oil determines the Emulsifying Stability (ES) and Emulsifying Activity (EA). ES of protein is related to the ability to minimize the interfacial tension between oil and water when in emulsion. Surface activity is determined through the ability of protein to migrate, deploy, absorb as well as rearrange at interface. The region of the minimum solubility of proteins (isoelectric region) was the region of least soluble and minimum emulsifying capacity. The emulsifying properties vary with the two effects (1) absorption of the protein at oil or water interface results in a substantial decrease in interfacial energy (2) structural, electrostatic and mechanical energy barriers are caused by the interfacial layer that opposes destabilization (*Chaparro Acuña, Gil González & Aristizábal Torres, 2012*; *Kumar et al., 2011*). This property makes both EA and ES opposite to each other. Higher EA of WF (>40%) makes it a preferable choice compared to SF.

*Akintayo, Oshodi & Esuoso (1999)* showed that Foaming Capacity (FC) is associated with the flexibility of protein molecules which reduces the surface tension as well as the globular protein which can hinder surface denaturation, therefore, leading to a low FC (<9% in case of SF). The foaming capacity indicates that proteins have active sites on the flour. Soluble proteins reduce the surface tension when at interface between the fluid surrounding the molecules and air bubbles, which blocks the coalescence. Protein molecules can also be deployed, which interact with each other to give multilayer or film protein which increases the flexibility of air-liquid interface. This results in harder foam due to unbreakable bubbles (*Adebowale & Lawal, 2003*). Higher FC and Foaming stability (FS) of WF indicates that protein structures are not denatured yet and they still carry the capacity; however, in case of SF they seem to have lost their arrangement due to which reduced foaming properties has been observed.

Protein gelation is very important in several vegetables and other food items. The effective overlapping of the functional groups between adjacent protein moieties is very important for the gel network formation. Higher gelation capacity (Gelatinization temperature) for WF indicates that this may not be a good choice in bakery items.

The flour at the beginning of the process comes from the endosperm, which is rich in starch and as it reaches the end of grinding, flour comes from periphery which is rich in ash, pentosan and protein. During the milling process, the seeds pass through heavy grinders to attain fine powdered flour. This flour is further sieved and final product, obtained after series of sieving and treatment with BP, is packed and sold as commercial white flour used for bakery items. Most of the components such as crude protein, gluten, damaged starch datasets showed highly significant difference ($P < 0.001$) with the WF dataset. This shows that quality of SF was deteriorated during processing and refining.

Flour mills consume benzoyle peroxide (BP) to improve the appearance and white color of flour. BP is a free radical initiator and therefore, it causes the oxidation of carotenoids by free radical mechanism. The process (Fig. 3) leads to the formation of benzoic acid (BA) as a by-product (*Saiz, Manrique & Fritz, 2001*; *Shan et al., 2007*; *Sumnu & Sahin, 2008*). *Onishi et al. (2004)* reported that Chigasaki Health Centre in Kanagawa Prefecture in Japan observed about 60–100 µg/g of BA in December 1999, which was found to be due to BP introduced in the food items; therefore BP was being

decomposed into BA. Current study suggested that not all BP was converted into BA, therefore traces are still left in the sample which was in compliance with the findings of the previous studies (*Onishi et al., 2004*; *Ponhong et al., 2015*). Further analyses of SF indicated that all samples had traces of BP and excessive amount of BA in them. Study by *Ponhong et al. (2015)* indicated that not all BP could get converted to BA and slight amount of BA is also introduced during the bleaching process that helps in the initiation of conversion process. The Joint FAO/WHO Expert Committee on Food Additives (JECFA) has approved the quantity of BP allowable to be about 0–40 mg/kg which has also been approved by *World Health Organization (2001)* considering the requirement of whitening of flour. However, increased temperature during baking also speeds up the processes of metabolism of BP to it's by products (and BA in particular). The maximum allowable level for BA is 150 mg/L according to European community food safety regulations (*European Commission, 1995*), which means daily intake of 55.8 mg of BA per person or 0.8 mg/kg body weight assuming 70 kg of weight is allowed (*Vandevijvere et al., 2009*). According to JECFA acceptable daily intake was 0–5 mg/kg for BA and benzoate. Intake of benzoic acid in sensitive persons, lower than 5 mg/kg of body weight per day has been observed to cause non-immunological contact reaction. A few studies have reported strong allergic reactions such as urticarial, pruritus and rhinitis to benzoic acid and benzoate exposure. According to current study, Table 5, SF consumers gain 44.3 mg/kg/bw of BA per day which is above maximum allowable intake (5 mg/kg/bw) compared to those who consume WF.

Wheat flour is an unavoidable commodity. This shows that white SF as produced out of milling process are contributors of excessive amount of toxic benzoic acid (BA) in the consumers' body. Results of current study indicated very high amount of BA entering in human body upon consumption of SF. In China standard limit of BPO is not exceed 60 mg/kg; the maximum content of BP in wheat flour is 80 mg/kg in the US, in Japan 300 mg/kg, and also 50 mg/kg in United Kingdom. In China, permissible amount of BPO are 0.045, 0.05 0.06 g/kg according to standards of food additives regulation (*Wei, Zhujun & Liu, 2006*). According to Japanese regulations allow the use of diluted BPO (19–22% w/w) in wheat flour which is lesser than 0.30 g/kg. In France, the use of BP is strictly banned. In UK and USA the permitted level BPO are 0.05 g/kg, 0.045 g/kg respectively (*Saiz, Manrique & Fritz, 2001*). These standard limits demonstrated that the concentration of BP in SF was within permissible limit, however when this value was calculated on the basis of Avg. Daily Intake (ADI) and Estimated Daily Intake (EDI) (Table 5) it was observed that influx of BA in the body of consumers is very high, which is 44.3 mg/kg/day compared to allowable level of 0–5 mg/kg/day. Current study showed an EDI of 78.3 g/day if a person consumes Results revealed that regular consumption of SF has profound effect on human health, therefore WF should be a preferable choice instead.

# CONCLUSIONS

Wheat flour is one of the most important ingredients of food being consumed most frequently. To improve the baking quality, SF is often used instead of WF. Increased

demands of fine texture and bleached color has led flour millers to add enhanced concentration of BPO and extensive milling along with sieving. Extensive milling and increased BPO reduce the nutritive value of SF and enrich it with the toxicant such as BA (as degradation product of BPO). Among various parameters of flour quality, protein and ash content in WF sample was more than SF samples. Therefore, there is need to improve wheat flour quality being sold in the market by limiting the rate of BPO added as bleaching agent as well milling process should be improved. HPLC analyses effectively demonstrated the dissociation of BPO to BA, which means that BA in SF was due to BPO added as bleaching agent.

### Funding
This work was supported by Key R&D Programs in Hainan Province of China (#ZDYD2019194). The funders had no role in study design, data collection and analysis, decision to publish, or preparation of the manuscript.

### Grant Disclosures
The following grant information was disclosed by the authors:
Key R&D Programs in Hainan Province of China: #ZDYD2019194.

### Competing Interests
The authors declare that they have no competing interests.

### Author Contributions
- Numrah Nisar conceived and designed the experiments, performed the experiments, analyzed the data, prepared figures and/or tables, authored or reviewed drafts of the paper, and approved the final draft.
- Faiza Mustafa conceived and designed the experiments, prepared figures and/or tables, and approved the final draft.
- Arifa Tahir performed the experiments, prepared figures and/or tables, and approved the final draft.
- Rashad Qadri analyzed the data, authored or reviewed drafts of the paper, and approved the final draft.
- Yaodong Yang conceived and designed the experiments, authored or reviewed drafts of the paper, and approved the final draft.
- Muhammad Imran Khan conceived and designed the experiments, authored or reviewed drafts of the paper, and approved the final draft.
- Fuyou Wang performed the experiments, authored or reviewed drafts of the paper, and approved the final draft.

### Data Availability
Raw data is available in the Supplemental Files.

## Supplemental Information

Supplemental information for this article can be found online at http://dx.doi.org/10.7717/peerj.8788#supplemental-information.

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
