# Peer review of "Proximate composition, functional properties and quantitative analysis of benzoyl peroxide and benzoic acid in wheat flour samples: effect on wheat flour quality"

_PeerJ, doi:10.7717/peerj.8788_

## Round 0.1 · original submission · Major Revisions

Dear authors,

Reviewers have now commented on your paper. You will see that they are advising that you revise your manuscript. If you are prepared to undertake the work required, I would be pleased to reconsider my decision. Please, can I also ask you also to carefully revise the English of the manuscript.

Please, the quality of tables and Figures need better quality, and I ask you to accordingly check this point.

Kind regards
Dr. Mohammed Gagaoua

·

Basic reporting

Comment on language
Authors must check the English language and ensure that the paper is clearly written in standard, scientific English language, to make text easy to understand.
Example: lines 57-59 this paragraph is unclear, please consider rewriting, and how yellow color of WF can affect the dough??

suggestion:
All abbreviations must be clear in the first time in introduction
Line 94 : Please specify the genus and the species of wheat in the sample collection section even it is clear

Please change ml by mL in whole manuscript

Line 111 : Please change : described by Beuchat

Line 116 : check the form of reference : by Neto et al. (2001), the same in line 125 please check the form of all references as journal requirements.

Please check subscripts and superscripts in all calculation equation

Please include at the bottom of the Tables all the abbreviations included in the Tables

Authors should add the signifiance difference in all tables

Experimental design

Methods are well described with sufficient detail in manuscript and with high technical standard

suggestion:
Authors were used samples from local supplier with additives and a control sample without additives under optimized field conditions. If WF is the control please specify in background.
Authors must use the same sample (seeds grown in the same conditions) with and without additives to estimate the effect of BP and BA.
The proximate composition included gluten, protein, fat, fiber was lesser for SF then WF but these results can be attributed to the different samples and not only to the traitement by BP and BA. Same comment for foaming capacity, gelation capacity and other analysis used.
How authors explain the choice of samples (control and samples with additives).

Validity of the findings

no comment on finding

Conclusion : lines 419-421 : repeated idea in introduction. The reviewer suggests that the authors conclude the current result importance on the food science and/or consumer’s health, as well as their future projection.

Additional comments

The authors evaluated nutritive composition and concentration of benzoyl peroxide and benzoic acid in the wheat flour samples. Apparently, there are no studies on the limite of BP and BA in wheat flour. Therefore, the results revealed superiority of WF over SF in nutritive qualities and this is clear.

The manuscript is very concrete and with very complete information. Such study is important in the field of cereal science. Complete the study by estimate the dietary intake for BP and BA is very interesting and added more value to the manuscript. However, the reviewer wants to make some suggestions to improve its comprehension (cited above).

Reviewer 2 ·

Basic reporting

Line 41. Please add abbreviation (BA) after “Benzoic Acid” as first use
Line 75. … into BA, which is….
Line 81. Please put the tow references in the same brackets

Experimental design

Line 82-83. Explain why WF is not processed in the same way as SF, it would be highly interesting if the flours would have comparable treatments.
Please specify in “Sample collection” section that WF was whole wheat flour. Was the particle size of SF and WF characterized?
Line 117. Some instead certain
Line 118. Delete point after “min”, this is valid throughout the manuscript. Please specify the type of the used oil.
Line 131. Elsewhere, use h for hours (SI unit)
Line 141-142. Please give more details about crude fiber and fat determination
Line 189. Please add reference for Extraction Procedure for BPO and BA.
Statistics: you mention that the analysis were performed in triplicates. What is about the experimental setup, have the samples also been prepared in triplicates? Please indicate the significance level.

Validity of the findings

Authors state in materials and methods that Bulk density (BD) was measured and related results were showed in table 1, but not discussed.
Line 238. FC and FS of WF are more.
Table 1. Do not use % as a concentration unit - replace here by e.g. g/100 g or g/100 mL. Please use symbol “ρ” for bulk density instead BD.
Table 2. Please explain how gluten content was higher than crude protein content, knowing that gluten is an insoluble part of proteins.
Table 4. WF not WW

Additional comments

Manuscript titled " Proximate composition and quantitative analysis of benzoyl peroxide and benzoic acid in the wheat flour samples: wheat flour quality " investigates nutritive quality of commercial wheat flour (soft flour) by quantification of benzoyl peroxide added as bleaching agent in the SF, their proximate composition and functional properties.
There are many publications about benzoyl peroxide and benzoic acid in the wheat flour and their determination by HPLC, but topic is interesting and relevant for the field.

Reviewer 3 ·

Basic reporting

The English language should be improved to ensure that an international audience can clearly understand the text. Some examples where the language could be improved include lines 44-46, 66, 78, 229-231, 251-253, 283, 338, 361, 386, 407, 408, 410.

Experimental design

The authors should pprovide more details in some methods (lines 139 and 141-142).
More details about statistical analyses should be provided.

Validity of the findings

The authors should state their conclusion about the dietry intake of BP and BA.

Additional comments

Lines1-3. We suggest to choice an other title for the manuscript.
Line 32. Please write “The objective”.
Line 37 and line 95. The authors write “commercial soft flour samples”. Please delete the word samples to avoid redundancy.
Line 37-40. In this section, authors should report all the tests performed in the current study.
Line 110. Please write “Water absorption capacity “.
Line 110. Please write “(WAC and OAC respectively)”.
Line 116. Please write “Emulsifying stability and emulsifying activity (ES and EA respectively)”.
Lines 137-172. This section should be reconsidered to avoid redundancy.
Lines 173-204. WE suggest changing the title to be “BP and BA quantification”. Also in this section, authors should indicate the references.
Line 207. Please indicate the gender and the age of the 200 subjects.
Line 214. What is the meaning of wt?.
Line 215. Which post-hoc test was performed?
Line 276. Please write “flour” instead of “floor”.
Line 318. Please write “parameter” instead of “component”.
Line 375. Please write “during” instead of “while”.
Table 1 and table 2. Please add standard deviation and letters to clearly identify differences between mean values.
Table 4 and table 5. Please add letters to clearly identify differences between mean values.

---

## Round 0.2 · Minor Revisions

Dear authors,

Reviewers have now commented on your revised paper. You will see that one reviewer is still advising that you revise your manuscript, including English. If you are prepared to undertake the work required, I would be pleased to reconsider my decision.

Kind regards
Dr. Mohammed Gagaoua

Reviewer 2 ·

Basic reporting

no comment

Experimental design

no comment

Validity of the findings

no comment

Additional comments

no comment

Reviewer 3 ·

Basic reporting

The English language and style can be improved in the results section.

Experimental design

no comment

Validity of the findings

The conclusion should be reconsidered.

Additional comments

Lines 1-3. We suggest modifying the manuscript title to indicate the determination of functional properties of wheat flour in the current study.
Line 32. Please write “The objective” instead of “Objective”.
Line 37-44. In this section, authors should report all the tests performed in the current study (functional properties and proximate composition).
Line 46. Please delete the first sentence since it has been already reported in background section.
Line 64. Please write “the yellow colour”.
Line 95. Please delete “were”.
Line 114. Please delete “as”.
Lines 120-121. Please write “… were determined by the method described by Neto et al. (2001) with modifications”.
Line 130. Please write “the method reported by”.
Line 164. Please write “the method of”.
Line 183. Please write “cereal” instead of “clinical”.
Lines 199-200. Please rephrase this statement “About 8-10 g of sample in a sample bottle was added…”.
Line 206. Please write “BPO and BA were quantified in wheat flour as described by the method of Abe-Onishi et al. (2004).
Line 248. Please indicate the meaning of wt?.
Line 252. Please write “Tukey” instead of Turkey
Line 253. The authors reported a significant level of p <0.001 while others significant levels were reported in table 1 and table 2 (p < 0.05 and p < 0.01). Please indicate which significant level was used in statistical analysis.
Line 263. Please put “.” Instead of “:”.
Lines 264-339. The English language in the results section should be improved to ensure clear and professional English.
Line 400. Please write “Akintayo et al. (1999)”.
Line 451. Please write “mg/kg” instead od “mg.kg1”.
Lines 464-473. We suggest reconsidering the conclusion.
Table 1, 2, 3, 4 and 5. Please add letters to clearly identify differences between mean values.

---

## Round 0.3 · accepted · Accept

Dear authors,

I am glad to inform you that your manuscript is accepted for publication in PeerJ.

Kind regards
Dr. Mohammed Gagaoua